# Distribution of soil microorganisms in different complex soil layers in Mu Us sandy land

Zhen Guo[1,2]*, Haiou Zhang[1,2,3], Juan Li[1,2,3], Tianqing Chen[1,2,3,4,5], Huanyuan Wang[1,2,3,4,5], Yang Zhang[1,4]

1 Shaanxi Provincial Land Engineering Construction Group Co., Ltd., Xi'an, China, 2 Institute of Land Engineering and Technology, Shaanxi Provincial Land Engineering Construction Group Co., Ltd., Xi'an, China, 3 Key Laboratory of Degraded and Unused Land Consolidation Engineering, Ministry of Natural Resources, Xi'an, China, 4 Shaanxi Engineering Research Center of Land Consolidation, Xi'an, China, 5 Land Engineering Technology Innovation Center, Ministry of Natural Resources, Xi'an, China

* 675334047@qq.com

**Data Availability Statement:** All relevant data are within the manuscript and its Supporting Information files.

**Funding:** This study is financially supported by The Shaanxi Provincial Natural Science Basic Research

## Abstract

The soft rock in Mu Us Sandy Land has rich resources and high content of clay minerals. The combination of soft rock with sand can play a certain role in sand fixation and promote the green development of ecological environment. In this paper, the aeolian sandy soil in Mu Us Sandy was taken as the research object, and it was mixed with soft rock to form composite soil. The four volume ratios of soft rock to sand were respectively 0:1, 1:5, 1:2 and 1:1. And CK, P1, P2 and P3 were used to represent the above four volume ratios in turn. By means of quantitative fluorescent PCR and high throughput sequencing, 16S rRNA gene abundance and community structure were investigated. The results showed that the soil organic carbon (SOC) and total nitrogen (TN) contents in 0-30cm soil layer were higher. Compared with CK, the SOC of P2 was improved by 112.77% and that of P1 was 88.67%. The content of available phosphorus (AP) and available potassium (AK) was higher in 30-60cm soil layer, and P3 was more effective. The abundance of 16S rRNA gene in the mixed soil bacteria ranged from $0.03×10^9$ to $0.21×10^9$ copies $g^{-1}$ dry soil, which was consistent with the changes of nutrients. Under different soil layers, the three dominant bacteria in the mixed soil were the same, namely Phylum *Actinobacteriota*, Phylum *Proteobacteria* and Phylum *Chloroflexi*, and there were more unique genera in each soil layer. Both bacteria α and β diversity showed that the community structure of P1 and P3 in 0-30cm soil layers was similar, and that of P1 and P2 in 30-60cm soil layers was similar. AK, SOC, AN (ammonium nitrogen), TN and NN (nitrate nitrogen) were the main factors contributing to the differentiation of microbial community structure under different compound ratios and soil layers, and Phylum *Actinobacteria* has the largest correlation with nutrients. The results showed that the soft rock could improve the quality of sandy soil, and that the growth of microbial growth was dependent on the soil physicochemical characteristics. The results of this study will be helpful to the study of the microscopical theory for the control of the wind-blown sand and the ecology of the desert.

Program Project (HW, 2021JZ-57), Funded by
Technology Innovation Center for Land Engineering
and Human Settlements, Shaanxi Land Engineering
Construction Group Co.,Ltd and Xi'an Jiaotong
University (ZG, 2021WHZ0087), Shaanxi Province
Youth Science and Technology Nova Project (JL,
2021KJXX-88), Internal scientific research project
of Shaanxi Land Engineering Construction Group
(ZG, DJNY2022-24) and Innovation Capability
Support Program of Shaanxi (TC, 2021PT-053). ZG
conducted Conceptualization,Methodology,
Resources, Writing-original draft and Writing-
review & editing. HW conducted Funding
acquisition, Project administration and Supervision.
JL conducted Formal analysis and Resources. TC
conducted Investigation. The funders had no role in
study design, data collection and analysis, decision
to publish, or preparation of the manuscript.

**Competing interests:** The authors have declared
that no competing interests exist.

## Introduction

Soil bacteria are the most abundant group of soil microorganisms, accounting for about 80% of the total microorganisms, and they are also rich in functional types [1, 2]. Because of their high adaptability, small size, great quantity and big surface area, bacteria have become the biggest living surface in the soil, so they are the most active living element, and they are always exchanging with the surrounding matter [3, 4]. Their can decompose organic residues in the soil, participate in the transformation of soil nutrients, and are the key organisms in the material cycle and energy flow of the ecosystem [5]. Along with more and more human interference to the soil, for example, the changes of soil control, fertilizer application and planting pattern, it is found that the disturbance has great influence on the structure, variety, and even function of the soil microorganism community [2]. However, the change trend of bacterial community in the soil of Mu Us Sandy Land with development potential still needs to be further explored, which provides a theoretical basis for the increase of national cultivated land area and quality improvement.

Microbes are the most sensitive to environmental conditions. The quantity, components and activity of microbial in the course of improving the sand soil can be greatly influenced by the use of different improving methods [6]. A large number of domestic and foreign scholars have studied the effects of agricultural use patterns on the soil quality of sandy farmland based on different experimental areas, indicating that if there is a reasonable land use pattern, soil carbon and nitrogen storage can improve the microbial quality of the region [7, 8]. We believe that long-term land use changes soil bacterial community structure in specific ways. The migration and interaction between microbial communities in the early stage of sandy land consolidation provides a unique environment for the development of soil ecosystems [2]. Liu *et al.* found that conservation tillage and fine management of irrigated farmland were beneficial to soil environment improvement and ecosystem restoration in sandy land [9]. Su *et al.* showed that after the desert sandy land was reclaimed into farmland, the soil fertility and microbial community were significantly improved with the increase of reclamation years, but the soil fertility in this area was still at a low level [10]. Moreover, some researches have also been done on the improvement of sandy soil by artificial sand wall, which can enhance the quantity of microbes and the activity of urease [11].

The Mu Us Sandy Land lies in a semiarid region in northern China [12]. Because of the shortage of surface water resources, low vegetation cover, vulnerable to human activities, and severe soil erosion, the ecology of Mu Us Sandy Land is vulnerable [13]. He *et al.* used an engineering measure to improve the sandy land, indicating that soft rock was a loose rock widely distributed in the Mu us Sandy land, and its mixing with aeolian sandy soil can significantly improve the water and fertilizer retention capacity of the sandy land [14]. Moreover, it was thought that the soft rock will be as soft as mud when it comes into contact with water, which can improve the chemistry and physics properties of the sand and the production of crops, and also enhance the colloid content of sand [15]. Therefore, it is suggested that the application of soft rock to improve the wind-blown sand in Mu Us Sandy Land can not only improve the soil moisture and fertilizer, but also increase the acreage of arable land, and improve the production of the crop, and resolve the material need of improving the Mu Us Sandy Land, and keep the ecological environment sustainable [12, 16].

Under the same site conditions, soil microorganisms showed vertical variation with soil depth. Liu *et al.* believed that bacterial community composition in desert areas was highly stratified, and surface soil microorganisms were greatly disturbed [17]. Du *et al.* found that the number and diversity of microorganisms decreased with the deepening of soil depth, and there were common and specific microbial groups in each soil layer [18]. It can be seen that

the depth of soil layer has an obvious effect on microbial gradient distribution. The improvement of sandy land by soft rock is an engineering measure to organically reconstruct the sandy land, which has a layered structure. The study on the influence of soft rock on soil bacteria in sandy land is of great significance to reveal the response mechanism of underground microbial community to engineering measures and to study the improvement measures of soil quality in sandy land. However, previous studies on soft rock and sand compound soil mainly focused on the physical structure and chemical properties, while there are few reports on the differences in soil bacterial community structure and its driving factors during soil development. Therefore, our research objectives are to (1) clarify the soil (soft rock and sand compound soil) nutrient gradient changes after sandy land improvement, (2) reveal the layered structure of the bacterial community in the compound soil, and (3) elucidate soil factors that regulate bacterial community structure in compound soils.

## Materials and methods

### Overview of the test site

The experimental area of soft rock-sand composite soil was situated in Mu Us Sandy Land (E109°28′58″-109°30′10″, N38°27′53″-38°28′23″) in Yuyang District of Yulin City, which lied in the region of Northwestern Shaanxi. The experimental region is a typical middle temperate semiarid continental monsoon climate belt, which was characterized by irregular rainfall in time and space, dry weather, long winter and short summer, four distinct seasons and abundant sunlight. The average annual temperature was 8.1°C, and the average annual frost was 154 days, the average annual rainfall was 413.9 millimeters, and the precipitation was 60.9% in June-September. The average number of sunshine hours per year was 2879 hours, and the proportion of sunlight was 65%. The soil type of the project area was mostly sand.

### Experiment design

The experimental field was used to simulate the soil conditions of the mixture of soft gravel and sand in the Mu Us Desert. In the experiment field with a 5 m × 12 m = 60 m$^2$, the chosen proportion of soft stone to sand (0: 1, 1: 5, 1: 2, 1: 1) was used for 3 times. And CK, P1, P2 and P3 were used to represent the above four volume ratios in turn. The field trial was carried out on an annual basis, starting in the middle of April, and picking from the middle to the end of September. Artificial planting method was used all year round. During the farming years, the application of chemical fertilizers is used to promote the growth of crops and the accumulation of root exudates, and at the same time promote the metabolic activities of microorganisms, and promote the increase of the nutrient content of the compound soil of soft rock and sand. The experimental fertilizer were composed of urea, diammonium phosphate and potassium chloride. The quantity of fertilizer applied was N 300 kg ha$^{-1}$, P$_2$O$_5$ 375 kg ha$^{-1}$ and K$_2$O 180 kg ha$^{-1}$ per year.

### Soil sample collection

After the potatoes were harvested in September 2020, soil samples from 0 to 60 cm (0–30 cm, 30–60 cm) were taken from each plot. In every field, three mixed soil samples were taken, and all of them were collected and mixed with five points. The soil samples were separated into two sections with the removal of plant and animal residues. One of them was natural air dried and screened with a sample of 1 mm and 0. 149 mm, and the other one was kept in a freezer at -80°C for microbiological analysis.

## Determination of soil physical and chemical properties

Determination of soil organic carbon (SOC) by external heating of potassium dichromate [19]. Determination of total nitrogen (TN) by Kjeldahl digestion, the molybdate blue colorimetric method for determination of available phosphorus (AP), and atomic absorption spectrometry for determination of available potassium (AK) [19]. $NO_3^--N$ and $NH_4^+-N$ were extracted at a ratio of 10 g fresh soil to 100 mL 2 M KCl. After shaking for 1 h, the extracts were filtered and analyzed by continuous flow analytical system (San++ System, Skalar, Holland) for $NO_3^--N$ and $NH_4^+-N$ [20]. pH was measured using a pH meter (PHS-3E, INESA, China), and the soil-to-water ratio was 1:5 [21].

## Soil DNA extraction and sequencing

The E.Z.N.A. ® Soil DNA Kit (Omega, Inc., USA) was used to extract the entire DNA from the soil sample. Then, the concentration and purity of the DNA were measured with the Nitragon 2000 spectrophotometer, and the results were measured with 1% agarose gel electrophoresis. The PCR amplification was performed with V3-V4 region specific primers 338F (5′-AC TCCTACGGGAGGCAGCAG-3′) and 806R (5′-GGACTACHVGGGTWTCTAAT-3′) based on the total microorganism DNA in every soil specimen [22].

## Fluorescence quantitative PCR amplification

Fluorescent quantitative PCR was carried out with the same primers as the above-mentioned high throughput sequencing [23]. Amplification was performed with a fluorescent PCR apparatus (Applied Biosystems, USA). Three copies were established for each specimen, and the final genetic abundance was calculated according to the dry mass of the soil.

## Data processing and analysis

The experimental data was analyzed with SPSS 20.0 for variance analysis. The original reading was obtained by sequencing. The quality of the high-throughput sequencing raw data was checked by FastQC software, and the low-quality reading was filtered. Then, FLASH (V1.2.7) was used for assembly and QIME re-filtering. The 97% similarity was used as the basis to ensure that the most effective data was used for clustering into operational taxonomic units (OTU). Sample composition was performed by using QIME (version 1.9.1) software, obtain the data of the bacterial community composition and relative abundance of the sample at different taxonomic levels, and make the relative abundance maps of the species at the Phylum and Genus level. The dilution curve analysis of OTU was performed using QIME (version 1.9.1) and the species diversity index was calculated. The β diversity is based on the distance matrix between the two samples to reflect the differences between the samples. The principal component analysis (PCA) diagram of soil bacterial community structure was drawn using R language (version 3.3.1). The redundancy analysis (RDA) between bacterial community composition and environmental factors was conducted using Canoco. The Spearman correlation coefficient was used to analyze the correlation between environmental factors and species, and the Heatmap was drawn with the aid of R software. The correlation between species was analyzed by Network.

# Results and discussion

## Soil physical and chemical properties

The SOC content in 0–30 cm soil layer was higher in P1 and P2 treatments, and P2 significantly increased the SOC content by 59.74% compared with P1 treatment. The TN content in

**Table 1. Vertical distribution characteristics of soil physical and chemical properties under different compound ratio treatments.**

| Compound ratio | Soil layer | SOC (g kg$^{-1}$) | TN (g kg$^{-1}$) | AP (mg kg$^{-1}$) | AK (mg kg$^{-1}$) | NO$_3^-$-N (mg kg$^{-1}$) | NH$_4^+$-N (mg kg$^{-1}$) | pH |
|---|---|---|---|---|---|---|---|---|
| CK | 0–30 cm | 2.35±0.39 aB | 0.15±0.03 aB | 3.71±0.69 aA | 20.33±4.16 bC | 2.25±0.17 bB | 2.95±0.56 aAB | 9.09±2.41 aA |
| | 30–60 cm | 2.02±0.15 aA | 0.16±0.02 aA | 4.58±0.57 aB | 28.49±2.67 aB | 4.87±0.63 aC | 2.43±0.67 aC | 9.01±1.95 aA |
| P1 | 0–30 cm | 3.13±0.35 aB | 0.28±0.09 aA | 3.02±0.50 aA | 37.40±6.15 aA | 7.00±1.43 aA | 3.83±1.11 bA | 8.96±1.30 aA |
| | 30–60 cm | 2.22±0.48 bA | 0.14±0.01 bA | 2.53±0.33 aC | 37.89±5.55 aA | 5.46±0.16 bB | 4.62±0.91 aAB | 8.94±2.16 aA |
| P2 | 0–30 cm | 5.00±1.04 aA | 0.09±0.01 aB | 2.96±0.39 aA | 33.42±5.11 aA | 8.22±2.33 aA | 3.54±1.36 aA | 9.11±1.99 aA |
| | 30–60 cm | 2.63±0.06 bA | 0.13±0.02 aA | 3.62±0.58 aABC | 33.12±5.32 aA | 6.61±2.12 bAB | 3.18±0.58 aB | 8.90±0.56 aA |
| P3 | 0–30 cm | 3.11±0.08 aB | 0.13±0.02 aB | 4.81±1.32 aA | 27.03±2.33 bB | 9.98±2.34 aA | 3.50±0.89 bA | 8.99±2.01 aA |
| | 30–60 cm | 2.64±0.13 aA | 0.11±0.03 aA | 6.45±1.23 aA | 34.53±4.33 aA | 7.51±2.06 bA | 5.31±1.14 aA | 8.78±0.89 aA |
| F value | Compound ratio | 22.4990** | 16.6590** | 62.3850** | 30.7910** | 607.1020** | 51.1090** | 0.1520 |
| | Soil layer | 0.0030** | 5.8850* | 18.1010** | 15.8230** | 70.3040** | 14.8840** | 0.5100 |
| | Interaction | 104830** | 12.2450** | 7.7510** | 5.0820* | 163.0560** | 23.8850** | 0.0710 |

Notes: CK, the volume ratio of soft rock to sand is 0:1; P1, the volume ratio of soft rock to sand is 1:5; P2, the volume ratio of soft rock to sand is 1:2; P3, the volume ratio of soft rock to sand is 1:1. SOC stands for soil organic carbon; TN stands for soil total nitrogen; AP stands for available phosphorus; AK stands for available potassium; NO$_3^-$-N stands for nitrate nitrogen; NH$_4^+$-N stands for ammo-nium nitrogen. Mean ± Standard deviation, lowercase letters indicate significant differences at the 5% level between different soil layers in the same treatment, uppercase letters indicate significant differences at the 5% level between different treatments in the same soil layer

*stands for P<0.05

** stands for P<0.01.

the P1 treatment was higher in the 0–30 cm soil layer, and there was no gradient difference in the other treatments. This is because crop roots are mainly distributed in 0-30cm soil layer, and root residues and their exudates increase nutrient content [24, 25]. The AP content of all treatments had no significant difference in the soil layer, and the AP content of the 30–60 cm soil layer P3 treatment was significantly higher than that of other treatments. The AK content of the P3 and CK treatments was higher in the 30–60 cm soil layer, which increased by 27.75% and 40.14% than surface layer, respectively. Because the soil structure becomes loose, porous and cementitious with the increase of the proportion of soft rock, and all the layers below 30 cm are sand layers, soil nutrients migrate downward due to leaching [16]. At the same time, the AK content of P1 treatment was higher at 0–30 cm and 30–60 cm. The contents of NO$_3^-$-N and NH$_4^+$-N in the 0–30 cm soil layer had no significant difference among treatments P1, P2 and P3, but increased compared with those of CK treatments. In the 30–60 cm soil layer, the contents of NH$_4^+$-N and NH$_4^+$-N were higher in the P3 treatment, and the increase was significantly higher than that in the CK treatment. pH was not significantly different among all treatments and soil layers (Table 1 and S1 Table). The results of two-factor test showed that compound ratio, soil layer and their interaction all had significant effects on SOC, TN, AP, AK, NO$_3^-$-N and NH$_4^+$-N, indicating that they had synchronous effects on soil properties, but had no effect on pH. Kang *et al.* [26] showed that the spatial structure of soil and the thickness of different soil layers had significant effects on soil nutrients, which was similar to the results of this study.

## 16S rRNA gene abundance of soil bacteria

Using fluorescence real-time quantitative PCR analysis, the abundance of 16S rRNA genes of the four mixed soil bacteria was between 0.03×10$^9$–0.21×10$^9$ copies g$^{-1}$ dry soil (Fig 1). In the 0–30 cm soil layer, the bacterial gene copy number in the P1 treatment was the largest, which was significantly increased by 65.77%-512.56% compared with other treatments. In addition,

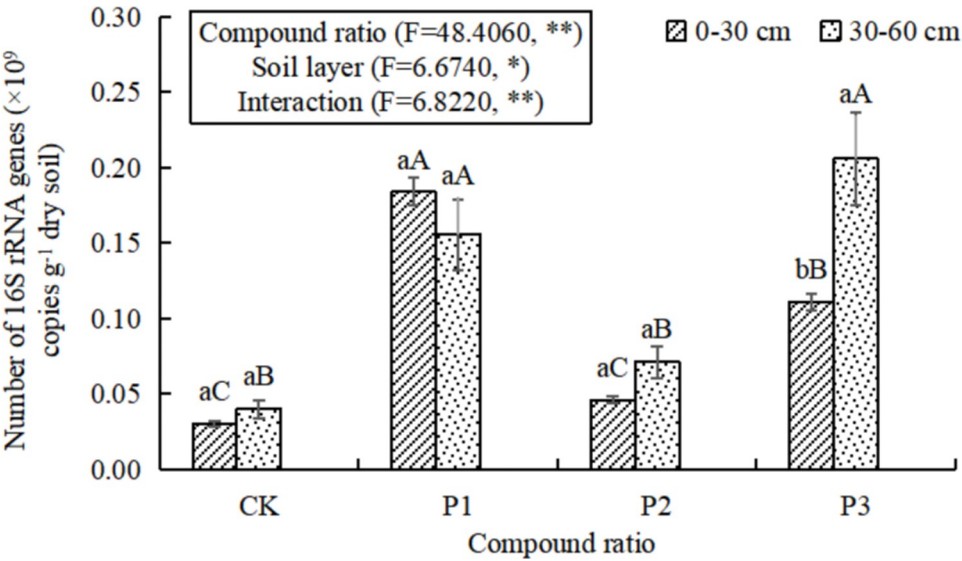

**Fig 1. The abundance of 16S rRNA of soil bacteria with different compound ratios.** Lowercase letters indicate significant differences between different soil layers under the same treatment (P<0.05), and capital letters indicate significant differences between different treatments on the same soil layer (P<0.05).

the gene copy number treated with P3 was larger, and there was no significant difference between P2 and CK treatments. In the 30–60 cm soil layer, there was no significant difference in gene copy number between P1 and P3 treatments, but it was significantly increased compared with CK and P2 treatments. The 30–60 cm soil layer showed a significant difference compared with the 0–30 cm soil layer only in the P3 treatment. The results of the two-factor test showed that the compound ratio, soil layer and their interaction had significant effects on the bacterial gene copy number, indicating that both the compound ratio and the thickness of the constructed soil layer had an important impact on the change of bacterial number.

## Bacterial community composition

According to the Phylum classification level, the bacteria community composition of the compound soil under different soil layers was studied, and the results showed the abundance of the top 12 bacteria. Others classified the relative abundance less than 0.01 into one category (Fig 2). The results of Steven *et al.* [27] showed that soil type did not affect the diversity of subsurface soil microbial communities. The results of this study showed that the three dominant bacteria were Phylum *Actinobacteriota*, Phylum *Proteobacteria*, and Phylum *Chloroflexi* in different soil layers. Among many wetlands, Phylum *Proteobacteria* has the highest relative abundance because of their strong adaptability to the environment [28]. The Phylum *Actinobaciota* has the highest abundance in this study, followed by Phylum *Proteobacteria*, indicating that Phylum *Proteobacteria* has high abundance in both dry land and wetland. Compared with CK, the compound ratio treatments in different soil layers increased the relative abundance of Phylum *Actinobacteriota*. In the 0–30 cm and 30–60 cm soil layers, the P1 and P3 treatments increased the relative abundance by 46.49% and 44.35%, respectively. Compared with CK, the relative abundance of Phylum *Proteobacteria* of compound soils under different soil layers showed a decreasing trend. The relative abundance of Phylum *Proteobacteria* under the soil layers of 0–30 cm and 30–60 cm all decreased significantly by P1 and P2. Compared with CK, the compound ratios both increased the abundance of Phylum *Chloroflexi* in the 0–30 cm and

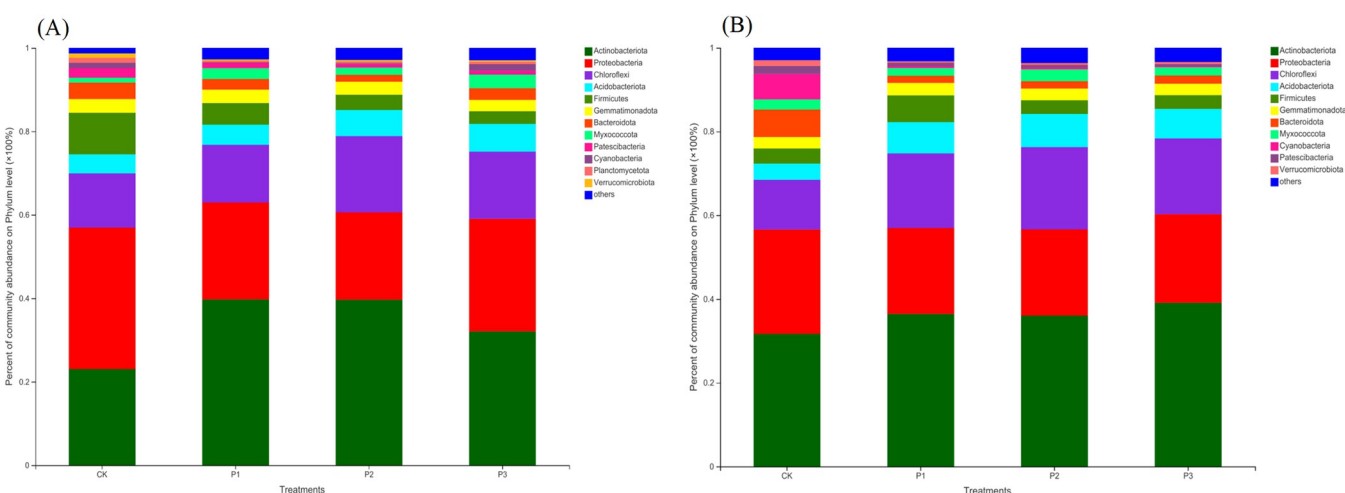

**Fig 2. Bacterial community composition based on Phylum level.** A is 0–30 soil layer, B is 30–60 cm soil layer.

30–60 cm soil layers, with the greatest increase in the P2 treatment, which were 47.26% and 58.39%, respectively. Compared with CK treatment, soft rock treatment reduced the relative abundance of non-dominant bacteria Phylum *Firmicutes*, Phylum *Bacteroidota*, Phylum *Cyanobacteria*, Phylum *Patescibacteria* and Phylum *Verrucomicrobiota*, but increased the abundance of Phylum *Acidobacteriota* and Phylum *Myxococcota*. It can be seen that at the level of Phylum classification, the bacterial species composition of soft rock and sand compound soil does not show obvious agglomeration at the vertical scale.

At the level of Genus classification, the differences of bacteria in different treatments increased and there were more endemic genera in each soil layer (Fig 3). In the 0–30 cm soil

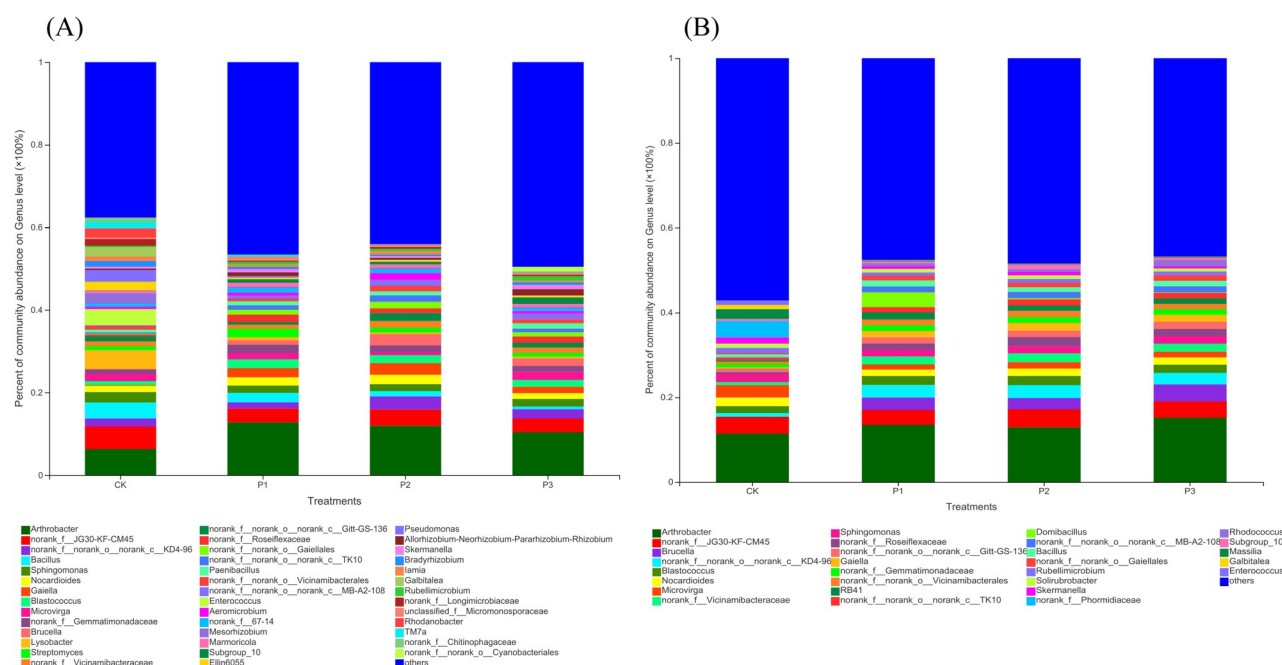

**Fig 3. Bacterial community composition based on Genus level.** A is 0–30 soil layer, B is 30–60 cm soil layer.

layer, the dominant bacteria of CK were Genu *Arthrobacter* (6.33%), Genu *norank_f__JG30-KF-CM45* (5.48%), and Genu *Lysobacter* (4.54%). The dominant bacteria of P1 were Genu *Arthrobacter* (12.75%), Genu *norank_f__JG30-KF-CM45* (3.39%), and Genu *Bacillus* (2.25%). The dominant bacteria of P2 were Genu *Arthrobacter* (11.93%), Genu *norank_f__JG30-KF-CM45* (3.96%), and Genu *norank_f__norank_o__norank_c__KD4-96* (3.18%). The dominant bacteria of P3 were Genu *Arthrobacter* (10.36%), Genu *norank_f__JG30-KF-CM45* (3.33%), and Genu *Bacillus* (2.25%). With the deepening of the soil layer, Genu *Arthrobacter* and Genu *Norank_F__JG30-KF-CM45* were still two dominant bacteria with high abundance, while other dominant bacteria were quite different. And in the 30–60 cm soil layer, Genu *Microvirga* (CK, 2.95%), Genu *Domibacillus* (P1, 3.50%), Genu *norank_f__norank_o__norank_c__KD4-96* (P2, 3.07%), and Genu *Brucella* (P3, 4.05%) were most abundant. This difference may be caused by the gradient difference in nutrient content of the mixed soil (Table 1 and S1 Table) or the adaptability of the mixed soil to the new environment is different [29, 30].

## Bacterial α diversity

Coverage refers to the sequencing accuracy of the sample library, and the higher the value, the higher the probability of the sequence in the sample being measured. The Coverage values in this study were all greater than 97%, indicating that the sequencing results were highly reliable and cover most of the sequencing information in the samples (Table 2). Chao and Ace indexes represent the abundance of bacterial communities, and the higher the value, the higher the abundance of community species. The results showed that the Chao index of different soil layers had no significant difference, and the Chao index of 0–30 cm soil layer was larger. In the 0–30 cm soil layer, the Chao index of P1 and P3 treatment increased significantly, while in the 30–60 cm soil layer, the Chao index of P1 and P2 treatment increased significantly. The change trend of Ace index was consistent with that of Chao index. Shannon index represented the diversity of bacterial community. The results showed that the addition of soft rock promoted the increase of bacterial diversity in sandy soil, but there was no significant difference between different treatments. Moreover, the increase of diversity in the surface layer was greater than that in the bottom layer. It may be due to the addition of soft rock clay minerals in the surface soil, which was greatly affected by the soil parent material [31].

## Bacterial community β diversity

PCA analysis results showed that CK was clearly distinguished from other processed samples on the PC1 axis, and other samples were located to the right of CK (Fig 4). In the 0–30 cm soil

**Table 2. Bacterial diversity of compound soil under different soil layers.**

| Soil layer | Compound ratio | Chao | Ace | Shannon | Coverage (%) |
|---|---|---|---|---|---|
| 0–30 cm | CK | 3282.61±21.63 bA | 3326.58±32.47 bA | 6.02±1.01 aA | 97.70 |
| | P1 | 4596.57±74.60 aA | 4640.46±80.90 aA | 6.32±0.74 aA | 97.08 |
| | P2 | 3690.49±101.25 abA | 3709.77±98.13 abA | 6.18±0.41 aA | 98.00 |
| | P3 | 4285.18±110.88 aA | 4301.08±101.56 aA | 6.36±0.82 aA | 97.51 |
| 30–60 cm | CK | 3114.89±90.20 bA | 3121.46±92.65 bA | 5.85±1.08 aA | 97.89 |
| | P1 | 4268.32±120.12 aA | 4269.10±110.88 aA | 6.13±1.24 aA | 97.87 |
| | P2 | 4361.71±105.50 aA | 4317.05±100.24 aA | 6.25±1.15 aA | 97.48 |
| | P3 | 3294.34±110.22 bAB | 3265.37±120.12 bAB | 5.89±1.01 aA | 98.01 |

Notes: Mean ± SD, Lowercase letters indicate significant differences between different compound ratios under the same soil layer (P<0.05), and capital letters indicate differences between different soil layers under the same compound ratio (P<0.05).

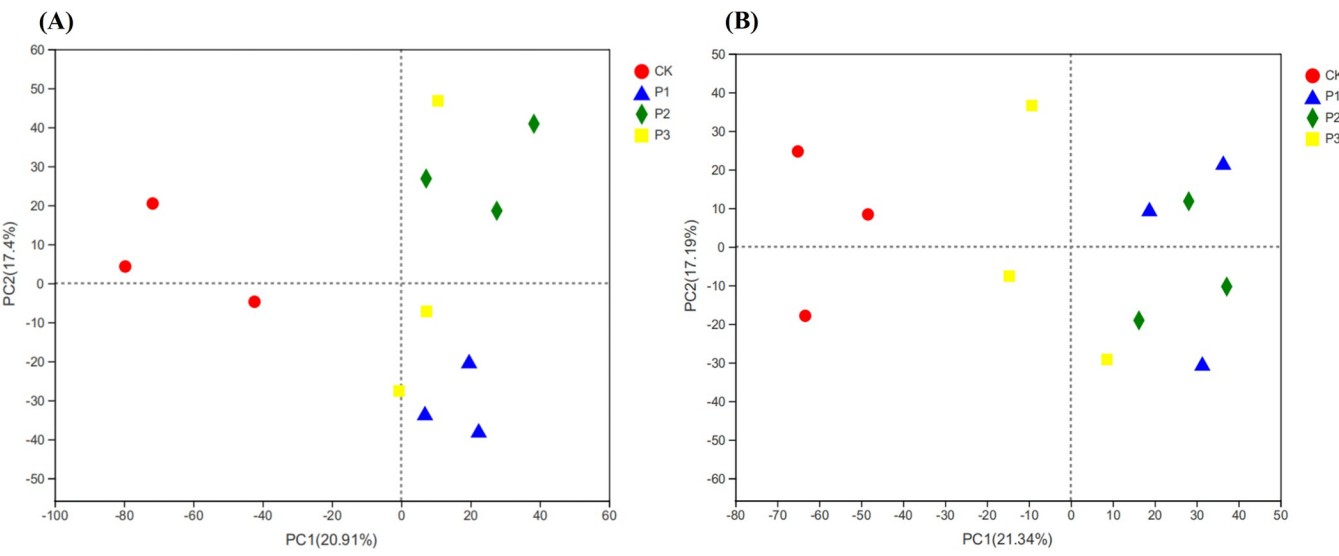

**Fig 4. The PCA analysis of bacterial community based on OTU level.** A is 0–30 soil layer, B is 30–60 cm soil layer.

layer, the PC1 and PC2 axes explained 20.91% and 17.40% of the total variation, respectively, and the distance between the P1 and P3 soil samples was relatively close (Fig 4A). In the 30–60 cm soil layer, PC1 and PC2 accounted for 21.34% and 17.19% of the total variation, respectively, where the small distance between soil samples P1 and P2 indicated similar bacterial community composition between them (Fig 4B). Among the four treatments, P1 and P3 had similar community structure in 0–30 cm soil layer, because the diversity and richness of bacteria also changed in the same way, and the abundance of Phylum *Myxococcota* in P1 and P3 treatments were higher than that in P2 treatments. In the 30–60 cm soil layer, the community structures of the P1 and P2 treatments were similar, because the species composition and abundance of the two treatments were not significantly different, resulting in a high degree of similarity between the two treatments. Bacteria have the highest diversity and the most stable community structure in medium-alkaline soil, but small changes in pH value may lead to the formation of different community structures [32, 33]. In this study, the pH values of P1 and P3 in 0–30 cm soil layer were basically the same, and the pH values of P1 and P2 in 30–60 cm soil layer were basically the same, showing the similarity of community structure.

## The relationship between soil properties and bacterial communities

In the 0–30 cm soil layer, the interpretation of the RDA1 axis and RDA2 axis was 54.08% and 14.75%, respectively, and the sum of the two axes was 68.83%. AK, SOC and AN had the greatest influence on bacterial community composition; NN, AP and TN were followed; pH had the least effect (Fig 5A). In the 30–60 cm soil layer, the interpretation of the RDA1 axis and RDA2 axis was 60.25% and 17.45%, respectively, and the sum of the two axes was 77.70%. The degree of influence of various environmental factors on the composition of bacterial communities in soil samples was AK, TN, and NN with the greatest impact; AN, SOC and pH had the second most impact; AP had the least impact (Fig 5B). However, the study of Kong *et al.* [26] was inconsistent with the results of this paper, arguing that pH was the main reason for affecting the bacterial community structure of surface soil, and ammonium nitrogen was the main reason for affecting the bacterial community of deep soil. This was because deep soil was more stable, while surface soil was susceptible to temperature, humidity and human activity. The

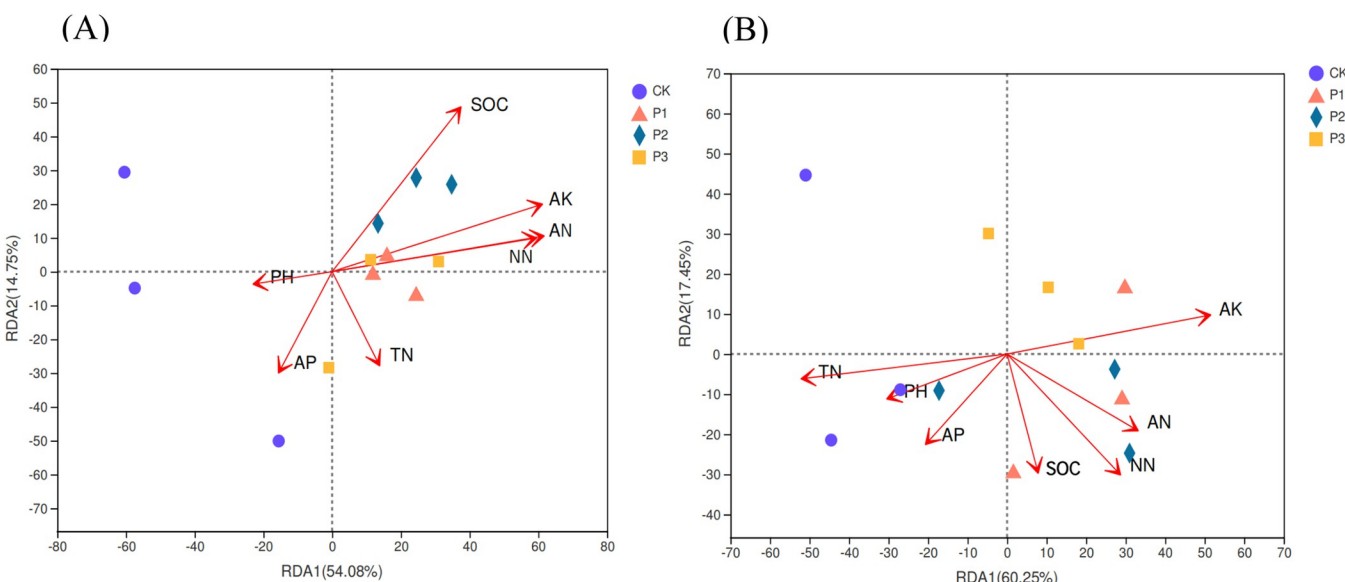

**Fig 5. Redundancy analysis (RDA) of bacterial community composition and soil chemical properties.** A is 0–30 soil layer, B is 30–60 cm soil layer. SOC stands for soil organic carbon; TN stands for soil total nitrogen; AP stands for available phosphorus; AK stands for available potassium; NN stands for $NO_3^-$-N; AN stands for $NH_4^+$-N.

results of RDA showed that AK content in different soil layers had a greater impact on the bacterial community composition of the samples, which confirmed that there was a synergistic change between soil properties and microbial communities in the process of sandy land improvement.

## Heat map of the correlation between soil properties and bacterial communities

The top 15 relative abundance species at the Genus level were selected for correlation analysis with soil properties. In the soil layer of 0–30 cm, soil properties had a great influence on bacterial community. Existing studies have suggested that the main nutrient sources of soil bacteria were root exudates and litters, and the quality and amount of nutrients provided by roots and litters for microorganisms were different, resulting in different soil bacterial community composition under different treatments [34, 35]. The Genu *Streptomyces* (belonging to Phylum *Actinobacteria*) was significantly negatively correlated with soil pH and positively correlated with TN and AK. The Genu *Gaiella* (belonging to Phylum *Actinobacteria*) was positively correlated with SOC and AN, and was significantly different from AK. The Genu *Norank_f__norank_o__norank_c__gitt-gs-136* was significantly positively correlated with SOC. The Genu *Arthrobacter* (belonging to Phylum *Actinobacteria*) and Genu *norank_f__Gemmatimonadanceae* were significantly positively correlated with AK. The Genu *Norank_f__Roseiflexaceac* showed significant positive correlation with NN and AN, and extremely significant positive correlation with AK. The Genu *Blastococcus* (Belonging to Phylum Actinobacteria) was significantly positively correlated with AN. The dominant bacterium The Genu *norank_f__norank_o__norank_c__kD4-96* had a significant positive correlation with SOC (Fig 6A). In the 30–60 cm soil layer, the Genu *Arthrobacter* (belonging to Phylum *Actinobacteria*) was positively correlated with AK, the Genu *Brucella* (belonging to Phylum *Proteobacteria*) was significantly negatively correlated with TN, and the Genu *Sphingomonas* (belonging to Phylum *Proteobacteria*) was significantly positively correlated with TN (Fig 6B). It can be seen that Phylum

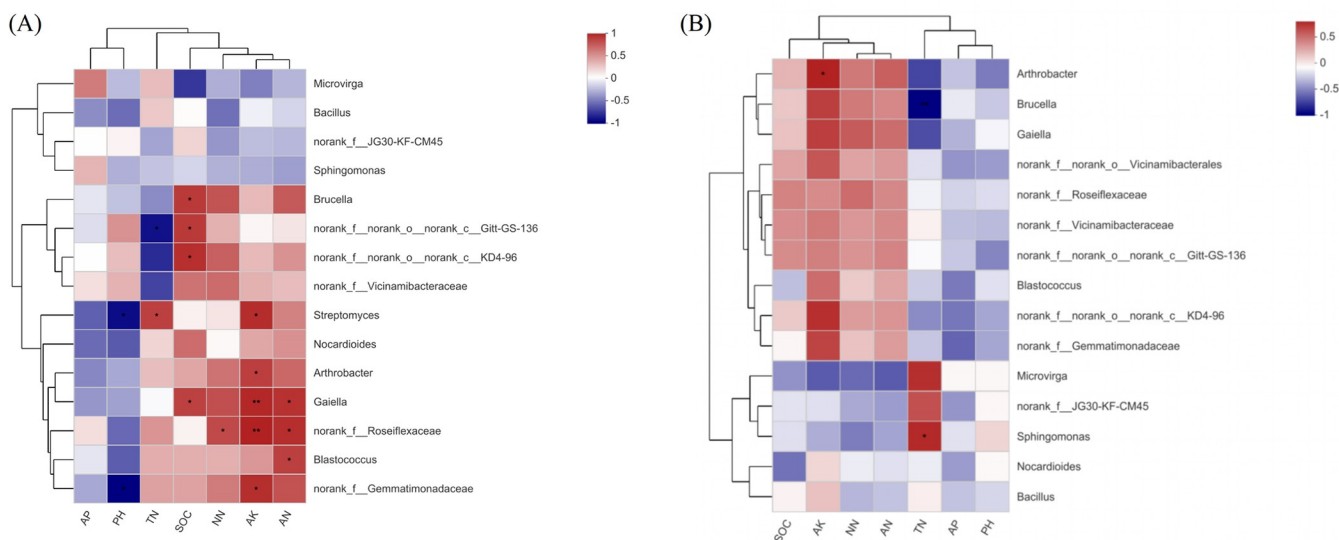

**Fig 6. A correlation heatmap of soil bacteria and soil chemical properties at the Genus level in different soil layers.** A is 0–30 soil layer, B is 30–60 cm soil layer. SOC stands for soil organic carbon; TN stands for soil total nitrogen; AP stands for available phosphorus; AK stands for available potassium; NN stands for $NO_3^-$-N; AN stands for $NH_4^+$-N. If the P value is less than 0.05, it is marked with *, and if the P value is less than 0.01, it is marked with **.

*Actinobacia* was the first dominant group in the composite soil, with high relative abundance in each soil layer, and was the main source of soil nutrient supply, which was related to that Phylum *Actinobacia* was suitable for growing in the neutral alkaline pH soil.

## Conclusions

The soil nutrient content and microbial diversity in Mu Us Sandy land can be increased effectively by combining soft rock with sand through land engineering measures. The compound ratio and different soil layers have significant differences in soil physical and chemical properties. The content of soil organic carbon and total nitrogen in the surface layer is higher, and the content of available phosphorus and available potassium in the bottom layer is higher. Under different soil layers, the three dominant bacteria in the mixed soil were the same, namely Phylum *Actinobacteriota*, Phylum *Proteobacteria* and Phylum *Chloroflexi*, which showed no obvious agglomeration on the vertical scale. Among them, Phylum *Actinobacteria* is the most closely related to soil nutrient supply. With the deepening of the soil layer, there are more endemic genera in the soil. The bacterial diversity and community structure are higher in 0-30cm treated with 1:5 and 1:1 compound soil, and higher in 30-60cm treated with 1:5 and 1:2 compound soil. Soil factor is the main factor driving the spatial distribution of soil microorganisms. Available potassium, organic carbon, ammonium nitrogen, total nitrogen and nitrate nitrogen are the main factors driving the differentiation of microbial community structure under different mixing ratios and soil layers. The results of this study provide practical significance for the reclamation of sandy land and the increase of cultivated land resources. The improvement of comprehensive properties of aeolian sandy soil will provide a good theoretical basis for the development of green agriculture and carbon emission reduction effect in the next step. Therefore, in the future, the author will continue to study the function and metabolism of microorganisms in sandy land, and carry out the isolation and identification of relevant carbon-fixing microorganisms.

## Supporting information

**S1 Table. Analysis of soft rock and sand compound soil physical and chemical properties.** (XLSX)

## Acknowledgments

We thank research staffs for their contributions to this work.

## Author Contributions

**Conceptualization:** Zhen Guo.

**Data curation:** Haiou Zhang.

**Formal analysis:** Juan Li.

**Funding acquisition:** Huanyuan Wang.

**Investigation:** Tianqing Chen.

**Methodology:** Zhen Guo, Yang Zhang.

**Project administration:** Huanyuan Wang.

**Resources:** Zhen Guo, Juan Li.

**Supervision:** Huanyuan Wang.

**Writing – original draft:** Zhen Guo.

**Writing – review & editing:** Zhen Guo.

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
