## [Decision Letter · Decision Letter 0]

10 Feb 2023

PONE-D-22-35140Distribution of soil microorganisms in different complex soil layers in Mu Us Sandy LandPLOS ONE

Dear Dr. Guo

Thank you for submitting your manuscript to PLOS ONE. After careful consideration, we feel that it has merit but does not fully meet PLOS ONE’s publication criteria as it currently stands. Therefore, we invite you to submit a revised version of the manuscript that addresses the points raised during the review process.

We look forward to receiving your revised manuscript.

Kind regards,

Tunira Bhadauria, Ph.D.

Academic Editor

PLOS ONE

Journal Requirements:

https://peerj.com/articles/13561/

In your revision ensure you cite all your sources (including your own works), and quote or rephrase any duplicated text outside the methods section. Further consideration is dependent on these concerns being addressed.

3. PLOS requires an ORCID iD for the corresponding author in Editorial Manager on papers submitted after December 6th, 2016. Please ensure that you have an ORCID iD and that it is validated in Editorial Manager. To do this, go to ‘Update my Information’ (in the upper left-hand corner of the main menu), and click on the Fetch/Validate link next to the ORCID field. This will take you to the ORCID site and allow you to create a new iD or authenticate a pre-existing iD in Editorial Manager. Please see the following video for instructions on linking an ORCID iD to your Editorial Manager account: https://www.youtube.com/watch?v=_xcclfuvtxQ.

Reviewers' comments:

Reviewer's Responses to Questions

**Comments to the Author**

1. Is the manuscript technically sound, and do the data support the conclusions?

Reviewer #1: Yes

Reviewer #2: Yes

2. Has the statistical analysis been performed appropriately and rigorously? 

Reviewer #1: Yes

Reviewer #2: I Don't Know

3. Have the authors made all data underlying the findings in their manuscript fully available?

Reviewer #1: Yes

Reviewer #2: Yes

4. Is the manuscript presented in an intelligible fashion and written in standard English?

Reviewer #1: Yes

Reviewer #2: Yes

5. Review Comments to the Author

Reviewer #1: Author are recoomended to improve manuscript

1."The soft rock in Mu Us Sandy Land has rich resources and high content of clay

minerals" In abstract pl write full form of Mu Us .

2."The four volume ratios of soft rock to sand were

respectively 0:1 (CK), 1:5 (P1), 1:2 (P2) and 1:1 (P3)."In abstract pl explain about CK,P1,P2.P3

3.I strogly recommend for publication of this work.soil parameter with meta genomics provide insight about soils are remarkable

4.Author are strongly suggested to pl write full farm of abbreviation at least once in manuscript so that reader feel easy.

5.Author are suggested to write future prospective of this work in conclusion.How this work might be helpful to increase diversity, conservation and crop production.

Reviewer #2: 1. The study was well-structured, yet some areas of the manuscript need attention.

2. Please correct the grammatical errors throughout the manuscript. Write in passive voice (Line no. 169 & Line no.173)

3. It will be Phylum Actinobacteria. The word Phylum must be written first, then the name of the Phylum.

4. There were observations but adequate discussions were missing.

5. Explain the observed relationships between soil properties and the bacterial abundances from previous literatures.

6. The significance of the study results was not reflected in the discussions/conclusions.

6. PLOS authors have the option to publish the peer review history of their article (what does this mean?). If published, this will include your full peer review and any attached files.

Reviewer #1: **Yes: **Kuldip Jayaswall

Reviewer #2: No

<quillbot-extension-portal></quillbot-extension-portal>

---

## [Author Response · Author response to Decision Letter 0]

18 Feb 2023

Response to Reviewers

Response: The author has modified according to the journal template requirements.

https://peerj.com/articles/13561/

In your revision ensure you cite all your sources (including your own works), and quote or rephrase any duplicated text outside the methods section. Further consideration is dependent on these concerns being addressed.

Response: The author rewrote the repeated text. References are cited. Thank you. 

3. PLOS requires an ORCID iD for the corresponding author in Editorial Manager on papers submitted after December 6th, 2016. Please ensure that you have an ORCID iD and that it is validated in Editorial Manager. To do this, go to ‘Update my Information’ (in the upper left-hand corner of the main menu), and click on the Fetch/Validate link next to the ORCID field. This will take you to the ORCID site and allow you to create a new iD or authenticate a pre-existing iD in Editorial Manager. Please see the following video for instructions on linking an ORCID iD to your Editorial Manager account: https://www.youtube.com/watch?v=_xcclfuvtxQ.

Response: The author provided ORID iD: 0000-0003-3383-6341

Reviewer #1: Author are recoomended to improve manuscript

1."The soft rock in Mu Us Sandy Land has rich resources and high content of clay minerals" In abstract pl write full form of Mu Us .

Response: "Mu Us Sandy Land" is an official name that refers to a specific place, and therefore does not exist in the full form of Mu Us.

2."The four volume ratios of soft rock to sand were respectively 0:1 (CK), 1:5 (P1), 1:2 (P2) and 1:1 (P3)."In abstract pl explain about CK,P1,P2.P3

Response: The four volume ratios of soft rock to sand were respectively 0:1, 1:5, 1:2 and 1:1. And CK, P1, P2 and P3 were used to represent the above four volume ratios in turn. 

3.I strogly recommend for publication of this work.soil parameter with meta genomics provide insight about soils are remarkable

Response: Thank you very much for your suggestion. Other questions have been modified.

4.Author are strongly suggested to pl write full farm of abbreviation at least once in manuscript so that reader feel easy.

Response: The author gives a complete explanation where the abbreviations first appear in the whole text. In addition, the author of each part makes a complete description of abbreviations. Thank you！

5.Author are suggested to write future prospective of this work in conclusion.How this work might be helpful to increase diversity, conservation and crop production.

Response: The author has added the future prospective of this work in conclusion. “The improvement of comprehensive properties of aeolian sandy soil will provide a good theoretical basis for the development of green agriculture and carbon emission reduction effect in the next step. Therefore, in the future, the author will continue to study the function and metabolism of microorganisms in sandy land, and carry out the isolation and identification of relevant carbon-fixing microorganisms.”

Reviewer #2: 

1. The study was well-structured, yet some areas of the manuscript need attention.

Response: The author revised the full text according to the comments. Thank you.

2. Please correct the grammatical errors throughout the manuscript. Write in passive voice (Line no. 169 & Line no.173)

Response: The author has modified the sentence in the passive voice.

3. It will be Phylum Actinobacteria. The word Phylum must be written first, then the name of the Phylum.

Response: The author has revised the whole text. Thank you.

4. There were observations but adequate discussions were missing.

Response: The author adds a detailed discussion to the research results. Details are as follows:

Kang et al. [26] showed that the spatial structure of soil and the thickness of different soil layers had significant effects on soil nutrients, which was similar to the results of this study.

Among many wetlands, Phylum Proteobacteria has the highest relative abundance because of their strong adaptability to the environment [28]. The Phylum Actinobaciota has the highest abundance in this study, followed by Phylum Proteobacteria, indicating that Phylum Proteobacteria has high abundance in both dry land and wetland. 

Moreover, the increase of diversity in the surface layer was greater than that in the bottom layer. It may be due to the addition of soft rock clay minerals in the surface soil, which was greatly affected by the soil parent material [31].

Bacteria have the highest diversity and the most stable community structure in medium-alkaline soil, but small changes in pH value may lead to the formation of different community structures [32]. In this study, the pH values of P1 and P3 in 0-30 cm soil layer were basically the same, and the pH values of P1 and P2 in 30-60 cm soil layer were basically the same, showing the similarity of community structure.

…… 

5. Explain the observed relationships between soil properties and the bacterial abundances from previous literatures.

Response: The author has added. “However, the study of Kong et al. [26] was inconsistent with the results of this paper, arguing that pH was the main reason for affecting the bacterial community structure of surface soil, and ammonium nitrogen was the main reason for affecting the bacterial community of deep soil. This was because deep soil was more stable, while surface soil was susceptible to temperature, humidity and human activity.”

6. The significance of the study results was not reflected in the discussions/conclusions.

Response: The authors add research significance to the conclusion. “The results of this study provide practical significance for the reclamation of sandy land and the increase of cultivated land resources.”

---

## [Editor Report · Decision Letter 1]

6 Mar 2023

Distribution of soil microorganisms in different complex soil layers in Mu Us Sandy Land

PONE-D-22-35140R1

Dear Dr. Guo

We’re pleased to inform you that your manuscript has been judged scientifically suitable for publication and will be formally accepted for publication once it meets all outstanding technical requirements.

Kind regards,

Tunira Bhadauria, Ph.D.

Academic Editor

PLOS ONE

Additional Editor Comments (optional):

Reviewers' comments:

<quillbot-extension-portal></quillbot-extension-portal>

---

## [Editor Report · Acceptance letter]

30 Mar 2023

PONE-D-22-35140R1 

Distribution of soil microorganisms in different complex soil layers in Mu Us Sandy Land 

Dear Dr. Guo:

I'm pleased to inform you that your manuscript has been deemed suitable for publication in PLOS ONE. Congratulations! Your manuscript is now with our production department. 

Kind regards, 

on behalf of

Dr. Tunira Bhadauria 

Academic Editor

PLOS ONE